# Rational Prescribing of Pancreatic Enzymes for Patients with Pancreatic Cancer

**DOI:** 10.3390/pharmacy12020047

**Published:** 2024-03-06

**Authors:** Mary Acelle G. Garcia, Syed Imam, Ursula K. Braun, Leanne K. Jackson

**Affiliations:** 1Rehabilitation & Extended Care Line, Section of Palliative Medicine, Michael E DeBakey Veteran Affairs Medical Center, Houston, TX 77030, USA; syed.imam@va.gov (S.I.); ursula.braun@va.gov (U.K.B.); leanne.jackson2@va.gov (L.K.J.); 2Department of Medicine, Section of Geriatric and Palliative Medicine, Baylor College of Medicine, Houston, TX 77030, USA

**Keywords:** pancreatic cancer, exocrine pancreatic insufficiency, pancreatic enzymes, palliative care, medication instruction

## Abstract

Most patients with pancreatic cancer at some point present with symptoms related to exocrine pancreatic insufficiency (EPI). These include diarrhea, abdominal bloating, indigestion, steatorrhea, weight loss, and anorexia. Even though up to 80% of pancreatic cancer patients eventually present with symptoms related to exocrine pancreatic insufficiency, only 21% are prescribed pancreatic enzyme replacement therapy (PERT). Its effectiveness is also highly dependent on its proper timing of administration, and patients must be thoroughly educated about this. The impact of symptoms of EPI can lead to poorer overall well-being. Pharmacists play a crucial role in properly educating patients on the correct use of pancreatic enzyme replacement therapy. PERT is a key strategy in managing the symptoms of EPI and can improve quality of life, which is a central focus in palliative care. This treatment is profoundly underutilized in the palliative care of these patients. The objective of this review is to discuss the pharmacology, pharmacokinetics, side effects, available evidence of the effectiveness of pancreatic enzyme use for patients with pancreatic cancer, and challenges, along with proposed solutions regarding its use.

## 1. Introduction

Every year, about 64,000 patients are newly diagnosed with pancreatic cancer in the United States alone. It is considered the fourth leading cause of cancer-related death for both sexes, while its 5-year survival rate is as low as 12% [1]. Most of the tumors arise from the ductal epithelium of the head of the pancreas, causing mechanical obstruction of the flow of pancreatic juice, resulting in exocrine pancreatic insufficiency (EPI) [2]. A multi-institutional series of 185 exocrine pancreatic cancer patients documented its most common signs and symptoms: asthenia (86%), anorexia (83%), weight loss (85%), abdominal pain (79%), jaundice (55%), pale or clay-colored stools (54%), nausea (51%), diarrhea (44%), vomiting (33%), and steatorrhea (25%), all of which can be exacerbated by EPI [3]. Patients with EPI often describe foul-smelling flatulence, bloating, and exacerbation of pain by eating. Fat malabsorption is common; thus, patients are likewise prone to having fat-soluble vitamin deficiencies. For example, night blindness due to vitamin A deficiency, metabolic bone disease (i.e., osteomalacia or osteoporosis) due to vitamin D deficiency, increased oxidative cell stress from lack of vitamin E, and bleeding disorders due to vitamin K deficiency [4].

Almost two-thirds of patients with pancreatic head tumors will develop EPI, which increases to 9 out of 10 over the course of the illness [5]. Contributing factors for developing EPI in pancreatic cancer include obstruction of the main pancreatic duct by the tumor itself, reduced cholecystokinin secretion, bicarbonate delivery, and loss of primary parenchyma, including from surgical resection and irradiation [6]. Significant nerve damage after lymph node dissection during pancreatic surgery can also lead to asynchrony between bile and enzyme delivery, and reduced stimulation of the pancreas. 

Diagnostic imaging studies such as computed tomography scans, magnetic resonance imaging and endoscopic ultrasound cannot precisely diagnose EPI. Individuals with severe EPI exhibit an increased fecal fat concentration. The fecal elastase test is the most sensitive and specific test of pancreatic function [7]. Fecal elastase is an enzyme product of pancreatic secretion, which remains stable throughout the gastrointestinal tract. Values of 100–200 micrograms/gram of stool signify mild to moderate pancreatic insufficiency, while less than 100 micrograms/gram of stool indicate severe EPI. However, watery diarrhea may lead to a false positive test result due to dilution of the fecal specimen. Crohn’s disease, bacterial overgrowth, lactose intolerance, giardiasis, cholestasis and other biliary diseases, colitis, celiac disease, and short bowel syndrome, which are all diseases that impact mucosal fatty acid uptake, can also cause abnormal values [8]. These conditions should be considered if a patient has a poor response to PERT, as concomitant gastrointestinal comorbidity can also occur. Nevertheless, testing for fecal elastase is not a pre-requisite in starting a PERT prescription because of the high prevalence of EPI among pancreatic cancer patients. Empirical treatment with PERT is recommended in patients suspected of having fat malabsorption, as characterized by abdominal bloating, steatorrhea, diarrhea, and weight loss [7].

A patient-reported outcome instrument was developed to assess EPI’s symptoms and its unfavorable impact on health-related quality of life [9]. A retrospective and non-randomized study of 66 patients with unresectable pancreatic cancer, who were receiving PERT with standard palliative care (PC) versus standard PC alone, showed that the median survival was longer (301 days vs. 89 days) for the former group [10]. Poorly controlled malabsorption symptoms of patients with pancreatic cancer can significantly impact their quality of life. The unpredictable occurrence of gastrointestinal symptoms can limit their travel and disrupt their desired activities and social engagements, leading to isolation and emotional distress [11].

Palliative medicine is an integral part and the mainstay of treatment of pancreatic cancer patients. EPI symptoms can be alleviated by PERT; however, this is not typically taught in palliative care. Additionally, the main problem with PERT is its underuse among patients with pancreatic cancer. In a retrospective study of 4554 patients with pancreatic cancer, only 21.7% were prescribed PERT when, in fact, the prevalence of EPI was high [12]. The objective of this article is to provide a review of PERT and how palliative care clinicians can use it to palliate symptoms related to pancreatic cancer. 

## 2. Normal Physiology

The pancreas secretes 1.5 L of pancreatic juice daily, which is rich with enzymes to digest fats, proteins, and carbohydrates. Both hormonal and neuronal mechanisms regulate the secretion of pancreatic juice. Two hormones that provide a negative feedback mechanism are secretin and cholecystokinin. Secretin is released from enteroendocrine cells in the small intestine. It stimulates the pancreas to release a bicarbonate-rich fluid, which is important to neutralize gastric acid as it enters the duodenum. Cholecystokinin stimulates the pancreatic acinar cells to release digestive enzymes through the vagal afferents. It is released in response to the presence of proteins and fats in the small intestine from ingested food [13].

## 3. Pharmacology

Common indications for PERT include pancreatic cancer, intraductal papillary mucinous neoplasms, premalignant mucinous cystic lesions, benign tumor of the pancreas, pancreatectomy, cystic fibrosis, and chronic pancreatitis [14,15]. Pancreatic ductal adenocarcinoma represents the predominant subtype of exocrine pancreatic cancer by over 90% and has an overall 5-year survival rate of only 8.5% [15,16]. Cystic fibrosis is the most common cause of EPI among children, which is detrimental as it can affect their growth patterns, attributed to nutrient malabsorption [17]. Conversely, chronic pancreatitis is the primary cause of EPI among adults [14]. Incremental inflammation of the pancreas related to chronic pancreatitis can lead to the deterioration of pancreatic parenchyma and subsequently pancreatic insufficiency, resulting in malnutrition, weight loss and steatorrhea [18].

PERT helps improve digestion and absorb nutrients. The capsules are not systemically active substances, as the enzymes are not directly absorbed from the gastrointestinal tract [19]. The capsule is enteric-coated to help resist its destruction when it reaches the acidic environment of the stomach. The enzymes are only released when the capsule reaches the duodenum, with a higher level of pH 5.5 [20]. Patients who benefit from PERT have a dramatically decreased fecal fat content [21]. The fundamental principle is that enough pancreatic enzyme should be mixed with food at an appropriate pH environment for the greatest enzymatic activity. The patients should strictly adhere to detailed instructions specific to PERT administration, as discussed further below. 

PERT can also be administered through enteral tubes. It can be mixed with nectar-thick fruit juice for easier delivery and to prevent clumping in the tube [22]. It can also be mixed with sodium bicarbonate to dissolve the enteric-coating and allow for improved activation in a higher pH environment [8]. 

## 4. Pharmacodynamics

The inability to digest fat completely leads to the major maldigestion or malabsorption problems. In clinical trials, the administration of PERT as a mixture of amylase, lipase, and protease showed a significant improvement in the coefficient of fat absorption and nitrogen absorption and was accompanied by increases in body weight and body mass index [17]. 

Up to 85% of pancreatic cancer patients are expected to have weight loss [3]. A small trial of the combined intervention of pancreatic enzyme therapy for 8 weeks with dietary counseling resulted in a 0.7 kg improvement in body weight compared to a 2.2 kg weight loss for those in the placebo group among patients with unresectable pancreatic head mass [23]. This may indicate that administration of PERT can prevent further weight loss, given active treatment against fat malabsorption. Another study showed improvement in the feeling of indigestion, light-colored yellow stools, and visible food particles in stool for those patients who took PERT appropriately [24,25]. One observational study involving PERT use among patients with unresectable pancreatic cancer found an improvement in the body mass index by 1.01 (versus 0.95 in the placebo group, *p* ≤ 0.001), even while receiving chemotherapy [8]. Symptoms of fat malabsorption do not become clinically evident until the secretion of lipase is less than 10% of normal levels, highlighting the substantial reserve capacity of the pancreas [26].

## 5. Mechanism of Action

PERT is derived from porcine pancreatic glands. The capsules contain a combination of lipases, proteases, and amylases that catalyze the hydrolysis of fats into glycerol and fatty acids, proteins into peptides and amino acids, and starch into dextrins and short-chain sugar, respectively. The duodenum has a more alkaline (basic) pH environment because of the secretion of bicarbonate by the pancreas and bile from the liver. The enteric-coating of PERT is specifically formulated to dissolve in the duodenum, resulting in the release of the enzymes [20].

## 6. Pharmacokinetics

PERT is minimally absorbed from the gastrointestinal tract; hence, it is not absorbed into the bloodstream in any significant amount and therefore is not systemically active. Its effects are confined to the intestinal lumen, where it helps break down the food into absorbable components. PERT is eliminated in the feces entirely. Since PERT acts locally in the gastrointestinal tract, its volume of distribution, protein-binding, metabolism, elimination half-life, and clearance are not relevant [14].

## 7. Dosing and Formulation

PERT dosing is based on lipase units. The starting dose is 500 lipase units per kilogram of body weight per meal. The dosage should vary based on the fat content of the diet, clinical symptoms, and severity of steatorrhea [14]. A reduction in steatorrhea of up to 15 g fat per day is observed when PERT of 25,000 to 40,000 IU of lipase per meal is supplemented. However, doses should largely be dependent on the severity of the disease and size of the meal [27]. Additionally, one half of the computed dose per patient is recommended to be administered with snacks. 

PERT drugs are not bioequivalent and are not interchangeable, as the dosage formulations differ in their concentrations of lipase, protease, and amylase, as presented in Table 1. The Food and Drug Administration (FDA) has approved a total of five enteric-coated formulations: CREON^®^, ZENPEP^®^, PANCREAZE^®^, PERTZYE^®^ and RELIZORB^®^. CREON^®^ is a commonly available pancreatic enzyme replacement therapy, and it was the first PERT to be approved by the FDA in 2009. Enteric-coated formulations are designed primarily for patients who still maintain normal gastric acid secretion. A microencapsulated formulation is crucial to prevent the enzyme from deactivation. Both PERTZYE^®^ and RELIZORB^®^ are designed for use in gastrostomy tubes. The RELIZORB^®^ device is a cartridge filled with polymeric beads containing lipase enzymes to hydrolyze the fats present in enteral formulas [14]. The device itself has no quantified lipase units. VIOKACE^®^ is a non-enteric-coated formulation that is thought to mix well with intragastric contents and can rapidly release lipase in the duodenum for fat digestion; however, it is only prescribed for adult patients also treated with a proton pump inhibitor (PPI). The acid suppression action of a PPI prevents denaturation of uncoated exogenous pancreatic enzymes (i.e., VIOKACE^®)^. Table 2 shows the recommended guidelines for PERT dosages across various age categories. 

## 8. Toxicity and Adverse Effects

PERT acts locally in the gastrointestinal tract and is not significantly absorbed into the bloodstream; hence, it has a lower risk of systemic side effects [14]. Throughout clinical trials and post-marketing surveillance of PERT, there have been no reports of overdose. Clinical overdose studies proved no effect on the lungs, pancreas, liver, and kidneys, but it can produce symptoms such as diarrhea and stomach upset. Carcinogenicity studies have not been performed; hence, it remains important to monitor for potential side effects [14].

There is no contraindication for starting PERT and there is no known drug–drug interaction. As PERT is not absorbed, an effect on fetal development or reproduction is not expected. It is not known whether this is excreted in milk, but it can be safely given to pregnant women (pregnancy category C). This is generally a well-tolerated product, although rare side effects include oral irritation, which is typically observed in those who retain the capsule in their mouth or chew the capsule, which is not advisable. An increase in serum uric acid is noted; therefore, caution is advised among those with concomitant gout or hyperuricemia. Since the enzyme is derived from porcine sources, patients with a pork allergy should be advised accordingly. Fibrosing colonopathy has been reported as a side effect of PERT, which is typically found in patients with cystic fibrosis who require higher doses of pancreatic enzymes [14]. Patients who have Muslim or Jewish backgrounds who may avoid pork products due to religious reasons should be counseled regarding the origin of PERT. However, some religious authorities have granted its use as there is no formal alternative available [31]. It is essential for healthcare providers to collaborate closely with the patients and to consider cultural and religious sensitivities when prescribing treatments. 

## 9. Drug Interactions

PERT can cause a decrease in the absorption of ferric sulfate, resulting in a reduced serum concentration and potentially a decrease in efficacy [14].

## 10. Food Interactions

Patients should be instructed concerning the following: drink plenty of fluids, take with fluids and food, and if swallowing the oral capsule is not tolerated, it can be sprinkled on acidic soft foods with a pH of 4 or less [14]. There is no specific contraindication or well-documented interaction reported in the literature between PERT and alcohol; however, chronic and excessive alcohol consumption can lead to pancreatic damage and worsen pancreatic insufficiency. 

## 11. Assessment of Response to Therapy

The effectiveness of PERT can be assessed through clinical indicators such as decreased fat in the stools, gain in body weight and improvement in stool consistency [24]. Clinical studies have shown improvement in the symptoms of EPI as early as five days of PERT intake [32]. Keeping a food diary when starting PERT is an excellent strategy to optimize digestion and response to therapy. It should document the specific food intake, dose of enzyme taken and relief of symptoms. 

## 12. Challenges and Proposed Solutions

Palliative care clinicians should recognize exocrine pancreatic insufficiency symptoms related to pancreatic cancer in order to prescribe PERT, given its underutilization. 

Incorrect timing of PERT administration is another barrier to use. Patients should be counselled to take PERT at the time of food consumption, as the medication degrades within 10 min of intake. A prospective, randomized, crossover study concluded that the group who took PERT at the time of meals and during meals showed better fat digestion compared to those who took it before meals and just after meals [33].

A study by Landers et al. showed that patients taking PERT understood its complex medication titration as well as its appropriate use, which was the opposite of the study conducted by Garcia et al. [34,35]. However, all the participants enrolled in the latter study were older adults (i.e., age 65 and above) wherein cognitive impairment and health literacy may be contributing factors to poor knowledge regarding its proper intake. 

An additional PERT capsule is required for larger meals or higher fat content meals. Inadequate improvement in EPI symptoms despite PERT intake might suggest the necessity of escalating its dosage. The patients might not be aware that nutritional or shake-style supplements and vitamin D (a fat-soluble vitamin) require concomitant PERT use. An instructional handout can be provided detailing the following: (1) “Gather pancreatic enzyme capsules before meals and snacks, (2) Take pancreatic enzyme capsules with cold or room temperature liquids with the first bite of food or nutritional drink (e.g., Ensure or Glucerna). Take the capsule whole. Do not crush or chew it, (3) Set timer for 10 min. If you are still eating after 10 min, take another capsule and repeat until you are finished with your meal or drink, (4) Notify your physician if you continue to experience abdominal bloating, cramping, burping, increased frequency of stools or if stools continue to appear oily (steatorrhea)” [34]. Additionally, creating a concise instruction printed on the pill bottle is important to convey the essential information clearly and succinctly. Pharmacists are available to answer questions and provide further clarification on medication instructions, offering additional support beyond what is written on the pill bottle [36]. 

An additional cause of under-prescribing PERT may be related to the financial burden for a supportive care intervention. An analysis of 2020 Medicare Part D plans noted that the expected out-of-pocket costs of optimally dosed PERT across the five formulations ranged from USD 853 to USD 1536 per month for those paying a deductible and coinsurance, to USD 527 to USD 1210 for refills made after meeting the deductible for a 30-day supply [37].

Patients with pancreatic cancer also warrant a nutritionist referral due to the significant anorexia and weight loss, as nutritional supplements may be helpful [38]. Each patient’s needs are unique, and a nutritionist can provide tailored dietary advice and modifications to manage the symptoms of EPI. A nutritionist can also suggest strategies to ensure adequate intake of PERT. Referring a patient with pancreatic cancer for dietary counseling is a key component of a multidisciplinary approach to patient care. 

Patients with pancreatic insufficiency were traditionally managed by limiting the amount of daily fat intake; however, this led to further restriction of fat-soluble vitamin intake. The fundamental change in approach was then to limit foods that are difficult to digest (i.e., legumes) and advise the patients to have more frequent but low-volume meals [19,39].

If there is an insufficient response to initial therapy, two options to trial are: (1) to add a proton pump inhibitor (PPI) to decrease gastric acidity (PERT is denatured in the gastric acid of the stomach), or (2) to increase the dose of enzyme units as the dose that is needed is probably higher [40]. The maximum dose is 2500 lipase units per kilogram of body weight per meal (or less than or equal to 10,000 lipase units/kg of body weight per day) [11]. Interestingly, a case report was published involving unmasking the symptoms of pancreatic insufficiency when a pancreatic cancer patient inadvertently discontinued his PPI. The patient was not initially presenting EPI-related symptoms; hence, he was not prescribed PERT, likely because of his chronic use of the PPI, which led to a decrease in bicarbonate production and secretion. Within a few weeks of restarting both the PPI and PERT, the patient’s malabsorption symptoms had resolved [41].

## 13. A Case Report Highlighting Challenges with Regard to PERT Prescription Is as Follows

A 65-year-old man was diagnosed with metastatic pancreatic cancer refractory to multiple lines of therapy. His most burdensome symptoms were abdominal bloating, steatorrhea, and diarrhea, which restricted his capacity to leave the house. He was then referred to palliative medicine due to his complex medical needs. His condition raised concerns of moderate protein–calorie malnutrition. Upon establishing care with a palliative care clinician, the patient was already taking hydromorphone as needed for pain and CREON^®^ 1206, containing 6000 USP of lipase, 19,000 USP of protease and 30,000 USP of amylase, to aid digestion. However, despite his prescriptions, his adherence to CREON^®^ was irregular, particularly in timing it with meals. The patient admitted forgetting to take his CREON^®^ capsule with meals as he inadvertently placed it in his pillbox, together with his routine medication regimen. He likewise revealed that when he dines out at a restaurant, he forgets to bring the capsule with him and rather takes it as soon as he gets home. The CREON^®^ dose was further increased and education regarding its correct administration was provided. Recognizing the need for specialized nutritional intervention, the palliative care team referred him to a nutritionist. The nutritionist conducted a thorough assessment of the patient’s dietary habits, emphasizing the importance of taking CREON^®^ with meals. The patient also began taking CREON^®^ with his nutritional supplement or meal replacement shake (i.e., Ensure), a practice he was previously unaware of. Gradually, with consistent use of CREON^®^ with meals and adherence to the tailored dietary plan, the patient reported a significant improvement in his symptoms and was able to taper off his hydromorphone usage. This eventually led to a positive effect on his overall well-being. The collaboration between palliative care, nutrition, and the patient himself proved crucial in managing his exocrine pancreatic insufficiency, highlighting the importance of an interdisciplinary strategy in complicated cases. 

Regrettably, as his pancreatic cancer advanced, he became unable to tolerate any oral intake. He was subsequently placed in a hospice or comfort care at home. Afterwards, the CREON^®^ was discontinued since he could no longer tolerate oral intake.

## 14. Conclusions

Patients with pancreatic cancer typically present with pain, vomiting, diarrhea, and steatorrhea, which heavily impact their quality of life, affecting their day-to-day functioning and restricting social relationships. PERT can improve gastrointestinal symptoms and survival; however, it is underutilized in palliative care. Suboptimal patient instruction with regard to PERT use may also lead to poor patient adherence. Pharmacists play a crucial role in the management of patients taking PERT as their responsibilities span various aspects of patient care, including medication management, adherence support and education. A combination of dietary counseling, comprehensive PERT education and regular medication reviews, especially for older adults, is an important consideration in ensuring a holistic approach to managing the symptoms of pancreatic cancer. It is essential to manage pancreatic cancer symptoms using PERT, as this approach can potentially minimize a patient’s opioid usage and use of anti-diarrheal medications, although further studies are warranted. 

## Figures and Tables

**Table 1 pharmacy-12-00047-t001:** Available dosage forms and strengths of PERT from the Food and Drug Administration. USP: United States Pharmacopeia.

Dosage Forms	Lipase	Protease	Amylase
“CREON^®^” is a delayed-release capsule containing enteric-coated minimicrospheres.			
“CREON 1203”^®^	3000 USP	9500 USP	15,000 USP
“CREON 1206”^®^	6000 USP	19,000 USP	30,000 USP
“CREON 1212”^®^	12,000 USP	38,000 USP	60,000 USP
“CREON 1224”^®^	24,000 USP	76,000 USP	120,000 USP
“CREON 1236”^®^	36,000 USP	114,000 USP	180,000 USP
“ZENPEP^®^” is a delayed-release capsule containing enteric-coated beads.			
“EURAND 5” or ZENPEP 5^®^	5000 USP	17,000 USP	27,000 USP
“EURAND 10” or ZENPEP 10^®^	10,000 USP	34,000 USP	55,000 USP
“EURAND 15” or ZENPEP 15^®^	15,000 USP	51,000 USP	82,000 USP
“EURAND 20” or ZENPEP 20^®^	20,000 USP	68,000 USP	109,000 USP
“PANCREAZE^®^” is a delayed-release capsule containing enteric-coated microtablets.			
“McNEIL, MT 4” or PANCREASE 4200^®^	4200 USP	10,000 USP	17,500 USP
“McNEIL, MT 10” or PANCREASE 10,500^®^	10,500 USP	25,000 USP	43,750 USP
“McNEIL, MT 16” or PANCREASE 16,800^®^	16,800 USP	40,000 USP	70,000 USP
“McNEIL, MT 20” or PANCREASE 21,000^®^	21,000 USP	37,000 USP	61,000 USP
“PERTZYE^®^” is a delayed-release capsule containing bicarbonate-buffered, enteric-coated microspheres.			
“DCI 4” or PERTZYE 4^®^	4000 USP	14,375 USP	15,125 USP
“DCI 8” or PERTZYE 8^®^	8000 USP	28,750 USP	30,250 USP
“DCI 16” or PERTZYE 16^®^	16,000 USP	57,500 USP	60,500 USP
“VIOKACE^®^” is a regular-release (non-enteric-coated) tablet.			
“VIO9111” or VIOKACE 10,440^®^	10,440 USP	39,150 USP	39,150 USP
“VIO9116” or VIOKACE 20,880^®^	20,880 USP	78,300 USP	78,300 USP

Note: It is important to acknowledge that the USP units may not be directly equivalent to the units of activity used in other countries. Different jurisdictions may have alternative standards, which may lead to potential discrepancies in dosing recommendations and therapeutic equivalences.

**Table 2 pharmacy-12-00047-t002:** The recommended guidelines for PERT dosages across various age categories [26,28,29,30].

Age Group	Units of Lipase
Infant	2000–4000 units per 120 mL formula or breastmilk
	Dosing can be challenging due to the varying degree of fat content in breastmilk or formula
Child aged less than 4 years	1000 units per kilogram per meal
	500 units per kilogram per snack
Child aged 4 and above	500 units per kilogram per meal
	250 units per kilogram per snack
Adult starting dose	50,000 units per meal
	25,000 units per snack
Adult maximum dose	150,000 units per meal
	70,000 units per snack

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
