# Peer review of "Rational Prescribing of Pancreatic Enzymes for Patients with Pancreatic Cancer"

_pharmacy, 2024, doi:10.3390/pharmacy12020047_

Round 1

Reviewer 1 Report

Comments and Suggestions for Authors

In this manuscript, Garcia et al present relevant and interesting facts about pancreatic enzymes. However the authors should address the following concerns prior to publication:

1. In Food Interaction Section please include informations about interaction between alchool and pancreatic enzymes (Line 211).

2. Missing citations for lines 190, 191, 217. Please add the references.

3. In Mechanism of Action Section maybe you include more details.

4. Lines 200-202, can you be more specific about the “higher doses”?

5. Line 217, after how long should the effectiveness of PERT be evaluated?

6. Please combine sections 6,7,8,9,10,11 and 12 into a single section called Pharmacokinetics because there are few informations.

Author Response

We would like to say thank to the reviewer’s insights and constructive feedback which have been instrumental in enhancing the quality of our work.

1. Added in line 204: 

There is no specific contraindication or well-documented interaction reported in the literature between PERT and alcohol, however chronic and excessive alcohol consumption can lead to pancreatic damage and worsen pancreatic insufficiency.

2. References have been added accordingly.

3. Mechanism of action has been revised into:

PERT is derived from porcine pancreatic glands. The capsules contain a combination of lipases, proteases, and amylases that catalyze the hydrolysis of fats to glycerol and fatty acids, proteins into peptides and amino acids, and starch into dextrins and short-chain sugar, respectively. The duodenum has a more alkaline (basic) pH environment because of the secretion of bicarbonate by the pancreas and bile from the liver. The enteric-coating of pancrelipase is specifically formulated to dissolve in the duodenum resulting to release of the enzymes [18].

4. Toxicity and overdose portion revised into:

Throughout clinical trials and post-marketing surveillance of pancrelipase, there have been no reports of overdose. Clinical overdose studies proved no effect on lungs, pancreas, liver, and kidneys but it can produce symptoms such as diarrhea and stomach upset. Carcinogenicity studies have not been performed hence it remains important to monitor for potential side effects [12].

5. Edited the paragraph into:

The effectiveness of PERT can be assessed through clinical indicators such as decreased fat in the stools, gain in body weight and improvement in stool consistency [23]. Clinical studies have shown improvement in symptoms of EPI as early as five days of PERT intake [31]. Keeping a food diary when starting pancrelipase is an excellent strategy to optimize digestion and response to therapy. It should document specific food intake, dose of enzyme taken and relief of symptoms.

6. Now combined in one section entitled, 'pharmacokinetics'.

Thank you.

Reviewer 2 Report

Comments and Suggestions for Authors

pharmacy-2871387, Rational Prescribing of Pancreatic Enzymes for Patients with Pancreatic Cancer

The paper under review sheds light on a critical aspect of pancreatic cancer management - the underutilization of pancreatic enzyme replacement therapy (PERT) in alleviating symptoms of exocrine pancreatic insufficiency (EPI) in palliative care settings.

The review lacks depth in its critical analysis. A more thorough examination of the existing literature (it has a low number of references) would enhance the paper's credibility and provide readers with a comprehensive understanding of this problem.

The paper mentions challenges associated with PERT utilization, such as cost considerations and patient education, it does not discuss the potential solutions or strategies to overcome these barriers. A more detailed discussion on addressing these challenges, particularly through multidisciplinary approaches involving pharmacists and other healthcare providers, would strengthen the paper's practical relevance.

Some sections are underdeveloped, like section 2, 5, and 6-12. The authors should improve them considerably. Add some more references. The sections 6 up to 12, as well as 15 to 17 are based on a single source. A graphical support can be added to improve the quality of the paper.

Table 1 should be formatted in mdpi’s style. The title of the table is wrong, because it presents more products, not only Creon. The measure units should be presented in the first row. Add the pharmaceutical presentation form (ex. Capsules). Do they have a normal release or delayed release? The table is not analysed in any form. The authors should make comments on the values presented, like their range, the pharmaceutical forms, and so on.

Are there different opinions in the field? Some controversy? Please detail and critically analyse the data.

The authors should improve the message of the review. What are the main ideas that the readers should take from this work? What this work adds new to the existing body of literature? Highlight these points in the abstract, introduction and conclusions.

Comments on the Quality of English Language

OK

Author Response

We would like to extend our gratitude to the reviewers for their invaluable time and meticulous attention to detail in reviewing our manuscript.

Palliative care providers see patients with pancreatic cancer who present with pancreatic insufficiency, and yet it is not common practice to monitor the use of PERT and adjust its dosage, hence the relevance of this paper as it is underutilized in the field of practice.

Additional references have been added to reflect overall feedback received from the reviewers.

The pharmacists role as part of an interdisciplinary team has been further explained on section 12 and conclusion.

Underdeveloped sections have been revised accordingly (highlighted on the uploaded manuscript).

Table 1 is corrected with additional information.

Thank you.

Reviewer 3 Report

Comments and Suggestions for Authors

Dear Authors

Thank you for very interesting manuscript covering actually, as you pointed, quite forgotten problem in oncological suportive care and palliative care.

I would like to indicate some problems with bothers me whatsoever:

1. In my opinion it would be better to put together the points from 6 to 12 in one chapter Pharmacokinetics- as there is a chapter 4. Pharmacodynamics

2. All marketed names of drugs should be removed and only chemical names should be used; Table 1 should be removed (as I understand this Table covers drugs registred and used in the USA, but not all used in other countries); chapter 13 should be re-written without marketed names of drugs

3. The Case Report (chapter 19) is doubling informations from chapter 18, and one thing got my attention. Authors are citing their another research (position 32) and referring to the incorrect timing of PERT administration are describing like two different patients (verses 224-226) while in Case Report almost the same description is used for one patient (v 285-288). Nonetheless chapter 19 is in my opinion redundant.

Author Response

We would like to say thank to the reviewer’s insights and constructive feedback which have been instrumental in enhancing the quality of our work.

  1. Now combined in one section entitled, 'pharmacokinetics'.
  2. All ‘pancrelipase’ terms have been removed from the article and replaced with ‘pancreatic enzyme replacement therapy/PERT’ instead to promote neutrality and universality of the drug. The authors suggest keeping the brand names to show the variety of drug formulations unique to a specific brand, as well as importance of using only two specific brands (i.e., PERTZYE® and RELIZORB®) for patients with gastrostomy tubes
  3. Redundant information removed from initial section entitled, ‘challenges and proposed solutions’

Thank you.

Reviewer 4 Report

Comments and Suggestions for Authors

This is a very interesting study regarding to the rational prescribing of pancreatic enzyme for patient with pancreatic cancer.

The presentation is clear, and gives sufficient information about the topic.

I have only a few suggestions:

-          Line 81: “the pancreas secrets 1.5 liters of pancreatic juice”…this amount is daily? Please clarify it

-          Points 6-12: please merge under "Pharmacokinetics "title

-          Title 1.: I think instead of USP next the numbers should be units !!

              These pancreatic enzymes are measured in units.. may be USP units: e.g. 3000 U, or 3000 USP units, like insulin

-          Line 173: it is mentioned the bioequivalence; can you introduce a supplementary table for explanation of bioequivalence? It would be representative.

I have no other comments; I would accept it after these modification.

Congratulation for the authors.

Author Response

We would like to say thank to the reviewer’s insights and constructive feedback which have been instrumental in enhancing the quality of our work.

-          Line 81: “the pancreas secrets 1.5 liters of pancreatic juice”…this amount is daily? Please clarify it (the word daily has been added to the sentence, thank you).

-  Points 6-12: please merge under "Pharmacokinetics "title (Now combined in one section entitled, 'pharmacokinetics')

-          Title 1.: I think instead of USP next the numbers should be units !! These pancreatic enzymes are measured in units.. may be USP units: e.g. 3000 U, or 3000 USP units, like insulin. Line 173: it is mentioned the bioequivalence; can you introduce a supplementary table for explanation of bioequivalence? It would be representative.

Thank you for ensuring the accuracy of our references. The authors have double-checked, and it is confirmed that it is labeled as USP instead of units. Below the table, we have added the following statement to avoid confusion, “It is important to acknowledge that the USP units may not be directly equivalent to the units of activity used in other countries. Different jurisdictions may have alternative standards, which may lead to potential discrepancies in dosing recommendations and therapeutic equivalences.”

Merged in a sentence for a much more concise message: “PERT are not bioequivalent and are not interchangeable, as the dosage formulations differ in their concentrations of lipase, protease, and amylase.”

Reviewer 5 Report

Comments and Suggestions for Authors

Good nutritional care improves outcomes and is critical for your quality of life. A normally functioning pancreas secretes about 8 cups of pancreatic juice into the duodenum, daily. This fluid contains pancreatic enzymes to help with digestion and bicarbonate to neutralize stomach acid as it enters the small intestine. Pancreatic insufficiency is the inability of the pancreas to secrete the enzymes needed for digestion. Having an insufficient amount of pancreatic enzymes is very common among people with pancreatic cancer. When the pancreas does not produce enough enzymes to break down food, pancreatic enzyme products are needed. Pancreatic cancer and surgery to remove the cancer may reduce the numer of enzymes that the pancreas makes. The cancer can also block the enzymes from getting to the first part of the small intestine, where they are needed for digestion. This means that food is not properly digested, and the nutrients in it are not absorbed. These problems can be managed by taking capsules that replace the enzymes your pancreas would normally make as a pancreatic enzyme replacement therapy (PERT). Furthermore, pancreatic cancer is an aggressive tumor characterized by rapid local growth and spread to the lymph nodes and liver. It is usually diagnosed at an advanced stage because it does not cause any disturbing symptoms at the beginning. In the light of this information, the presented manuscript concerns a very important issue and, additionally, can bring us much closer to a not fully understood problem. The manuscript was properly structured in accordance with the requirements of the journal. The manuscript is well written and discusses some really interesting points. Particularly important is a point that is missing in many manuscripts or textbooks about medicines - the exact method of administration, including times, meals, and frequency of administration. I would like to thank the Authors for adding this point, which, in my opinion, significantly increases the value of the manuscript. It is extremely important and interesting for pharmacists who often provide information to patients and constitute the last link in the treatment chain.

I suggest making only minor corrections that will significantly improve the value of the manuscript.

In the abstract, the sentence: „Pancrelipase is an expensive drug, and it is inactive if not taken correctly” should be deleted. Furthermore, the objective of review should be moved to the end of the abstract.

In my opinion, the Authors should explain the statement: „Fecal elastase test is the most sensitive and specific test of pancreatic function in more detail; however, testing is not a prerequisite in starting pancreatic enzyme replacement therapy (PERT) because of the high prevalence of EPI among pancreatic cancer patients [7]”. Did they mean that this therapy was introduced routinely for every patient? This statement is not clear to me.

In my opinion, the Authors should significantly reduce the number of points included in the manuscript. And so, I propose to place points 6, 7, 8, 9, 10, 11 and 12 in one called Pharmacokinetics.

In line 175, the Authors provide the name RELIZORB®, but this preparation is not included in Table 1. There is a preparation called Viokace®, which is not mentioned in the text. The Authors should explain these discrepancies.

In point 14, I propose to emphasize that pancrelipase acts locally in the gastrointestinal tract and it is not absorber in any significant amount, which is a reason for few side effects.

In my opinion, the Authors should emphasize more the role of the pharmacist in the rational use of pancreatic enzymes by patients.

Author Response

We would like to say thank to the reviewer’s insights and constructive feedback which have been instrumental in enhancing the quality of our work.

In the abstract, the sentence: „Pancrelipase is an expensive drug, and it is inactive if not taken correctly” should be deleted. Furthermore, the objective of review should be moved to the end of the abstract.

Removed the phrase about its cost. Added the following instead: “Its effectiveness is also highly dependent on its proper timing of administration, and patients must be thoroughly educated about this.”

In my opinion, the Authors should explain the statement: „Fecal elastase test is the most sensitive and specific test of pancreatic function in more detail; however, testing is not a prerequisite in starting pancreatic enzyme replacement therapy (PERT) because of the high prevalence of EPI among pancreatic cancer patients [7]”. Did they mean that this therapy was introduced routinely for every patient? This statement is not clear to me.

Thank you, the statement has been moved at the bottom of the paragraph (after explanation regarding fecal elastase). The excerpt includes, “Nevertheless, testing for fecal elastase is not a pre-requisite in starting a PERT prescription because of the high prevalence of EPI among pancreatic cancer patients. Empirical treatment with PERT is recommended in patients suspected of having fat malabsorption, as characterized by abdominal bloating, steatorrhea, diarrhea, and weight loss [7].

In my opinion, the Authors should significantly reduce the number of points included in the manuscript. And so, I propose to place points 6, 7, 8, 9, 10, 11 and 12 in one called Pharmacokinetics. (Now combined in one section entitled, 'pharmacokinetics')

In line 175, the Authors provide the name RELIZORB®, but this preparation is not included in Table 1. There is a preparation called Viokace®, which is not mentioned in the text. The Authors should explain these discrepancies.

Added information regarding RELIZORB as follows: “The RELIZORB® is a cartridge filled with polymeric beads containing lipase enzymes to hydrolyze fats present in enteral formulas [12].”

Description regarding Viokace: “VIOKACE® is a non-enteric coated formulation that is thought to mix well with intragastric contents and can rapidly release lipase in the duodenum for fat digestion, however it is only prescribed for adult patients also treated with a proton pump inhibitor (PPI). The acid suppression action of a PPI prevents denaturation of uncoated exogenous pancreatic enzyme (i.e., VIOKACE®).”

In point 14, I propose to emphasize that pancrelipase acts locally in the gastrointestinal tract and it is not absorber in any significant amount, which is a reason for few side effects.

Added in line 181: PERT acts locally in the gastrointestinal tract and is not significantly absorbed into the bloodstream hence it has a lower risk of systemic side effects [12].

In my opinion, the Authors should emphasize more the role of the pharmacist in the rational use of pancreatic enzymes by patients.

Added in section 12:

Additionally, creating a concise instruction printed on the pill bottle is important to convey the essential information clearly and succinctly. The pharmacists are available to answer questions and provide further clarification on medication instructions, offering additional support beyond what is written on the pill bottle [36].

Added in conclusion:

The pharmacists play a crucial role in the management of patients taking PERT as their responsibilities span various aspects of patient care including medication management, adherence support and education.

Round 2

Reviewer 1 Report

Comments and Suggestions for Authors

Reviewer 2 Report

Comments and Suggestions for Authors

The authors made the requested modifications so I agree with the publication.